# Diagnosing Systemic Disorders with AI Algorithms Based on Ocular Images

**DOI:** 10.3390/healthcare11121739

**Published:** 2023-06-13

**Authors:** Huimin Li, Jing Cao, Andrzej Grzybowski, Kai Jin, Lixia Lou, Juan Ye

**Affiliations:** 1Eye Center, The Second Affiliated Hospital School of Medicine Zhejiang University, Zhejiang Provincial Key Laboratory of Ophthalmology, Zhejiang Provincial Clinical Research Center for Eye Diseases, Zhejiang Provincial Engineering Institute on Eye Diseases, Hangzhou 310009, China; 3180101896@zju.edu.cn (H.L.); janecao@zju.edu.cn (J.C.); jinkai@zju.edu.cn (K.J.); 2Institute for Research in Ophthalmology, Foundation for Ophthalmology Development, 60-836 Poznan, Poland; ae.grzybowski@gmail.com

**Keywords:** artificial intelligence, deep learning, machine learning, systemic disease, ocular image, cardiovascular diseases, neurodegenerative diseases, chronic kidney disease

## Abstract

The advent of artificial intelligence (AI), especially the state-of-the-art deep learning frameworks, has begun a silent revolution in all medical subfields, including ophthalmology. Due to their specific microvascular and neural structures, the eyes are anatomically associated with the rest of the body. Hence, ocular image-based AI technology may be a useful alternative or additional screening strategy for systemic diseases, especially where resources are scarce. This review summarizes the current applications of AI related to the prediction of systemic diseases from multimodal ocular images, including cardiovascular diseases, dementia, chronic kidney diseases, and anemia. Finally, we also discuss the current predicaments and future directions of these applications.

## 1. Introduction

Nowadays, the rapid emergence of artificial intelligence (AI) has prompted a gradual revolution in clinical consultation and medical management [1,2]. AI, a branch of computer science, refers to the imitation of human-like intelligent behavior by computers with minimal human intervention [1]. AI-derived image analysis is capable of unveiling intrinsic features from complex data. Moreover, its achievements have caused concern for several specialized forms of medical imaging analysis, including analysis of radiological images [3], pathology slides [4,5,6], skin lesions [7,8], electrocardiograms [9], and retinal images. Of these, emerging AI image analysis technologies are well-suited for ophthalmology [2,10,11], which has a high degree of equipment dependency and uses many image modalities, including fundus images, slit lamp images, and optical coherence tomography (OCT). Hence, ophthalmic photography, as an accessible avenue for directly visualizing neuro vasculature, may be able to incorporate well-established AI techniques for the prediction of systemic diseases.

Traditional diagnosis methods for systemic diseases characterized by the involvement of multiple organs often depend on a variety of laboratory and radiological examinations. However, concerns about frequent radiational exposure and a conservative attitude toward invasive blood tests diminish compliance with voluntary screening, which can be further exacerbated by the scarcity of complicated medical equipment at primary medical institutions. Thus, given the anatomical connection between the eyes and the rest of the body [12,13], a variety of systemic diseases that trigger the deterioration of neurovascular structures, are expected to be predicted based on the eye manifestations. Furthermore, detecting signs of systemic disease via ocular image-based AI—a non-invasive and inexpensive alternative approach to traditional screening—is expected to be a beneficial supplement to medical care, and may even serve as a substitution for traditional screening strategies. Despite evidence of satisfactory performance reported, in high-stakes scenarios, the inherent “black box” restriction of AI, along with other practical issues, may prevent its large-scale translation into clinical application.

Here, we focus on major image modalities in the ophthalmic examination and provide a summary of their current application for the diagnosis of systemic circulatory, neurological, respiratory, and digestive diseases. The scheme used for this review is shown in Figure 1. Finally, we discuss the limitations and the prospects of this technology for clinical deployment.

Abbreviations: AI, artificial intelligence; OCT, optical coherence tomography; OCTA, optical coherence tomography angiography.

## 2. Materials and Methods

A search of the scientific literature was conducted on 1 March 2023, using the PubMed, Web of Science, and Embase databases. It was completed following the Preferred Reporting Items for Systematic Reviews and Meta-Analysis (PRISMA) 2020 guidelines. The flowchart of the screening process is demonstrated in Appendix A.

### 2.1. Inclusion and Exclusion Criteria

We included relevant articles using search terms in specific combinations: “artificial intelligence”, “deep learning”, “machine learning”, “neural network”, “systemic disease”, “image”, and “eye”. The detailed search strategies are shown in Appendix A. Relevant articles were extracted from the references of identified articles to expand the scope of our search. Primary relevant studies were included if they relied on ophthalmic images to predict or evaluate a non-ocular systemic disease (or indicators of such a disease) using AI. All reviewed papers were available in full text and published in English. Irrelevant and duplicate articles were eliminated by examining the title and abstract. Review articles, surveys, case reports, comments, guidelines, patents, and editorials were not included. This review is limited to articles published from 2013 onwards.

### 2.2. Data Extraction and Quality Assessment

Paper screening, data extraction, and assessment were conducted by H.L. and J.C. Information including the author, publication year, target systemic diseases, ocular imaging modality, model architecture, training/testing dataset (data sources and sample size), external validation, model task (prediction, identification, or both), outcome, output form, and the model result were extracted from the included studies. The identification task means identifying the current state of the target systemic disease from ocular images in a cross-sectional study, while the prediction task refers to predicting future events or incident risk in a longitudinal cohort. The outcome column shows the initial evaluation results of the models, such as direct diagnosis, laboratory indicators, or biological parameters.

To evaluate the risk of bias in the included studies, the Quality Assessment of Diagnostic Accuracy Studies 2 (QUADAS-2) [14] was applied. The results of the risk of bias evaluation using QUADAS-2 are shown in Appendix A. All included articles are summarized in Table 1, Table 2, Table 3 and Table 4, and they are presented in Figure 2 sorted by imaging modalities and target systemic diseases.

## 3. Results

### 3.1. Detection of Cardiocerebral Vascular Diseases and Risk Stratification

#### 3.1.1. Detection of Retinal Microvascular Morphological Parameters for Stratification

Certain visible changes in the retinal arterioles, such as vessel caliber [51,52,53], bi-furcation, or tortuosity [54], have been highly valuable for suggesting that altered arteriolar function may exist throughout the body [55].

Deep learning models designed by Cheung et al. [15] for the automatic measurement of retinal-vessel caliber in fundus photographs performed well. Their performance was comparable to expert graders in associations between measurements of retinal-vessel caliber, and cardiovascular disease (CVD) risk factors, including blood pressure, body-mass index, total cholesterol, and glycated-hemoglobin levels. This demonstrated that reduced retinal arteriolar caliber and increased retinal venular caliber were both independently associated with an increased risk of incident CVD events.

Zekavat et al. [16] leveraged deep learning to quantify geometric microvasculature indices, and thereby characterized phenome-wide clinical associations and genomic risk factors. They reported that low retinal vascular fractal dimension and density were significantly associated with a higher risk of incident mortality, hypertension, and congestive heart failure. This novel framework emphasized how deep learning of images can quantify an interpretable phenotype that can then be integrated into clinical metadata, genetic data, and biomarker datasets to predict risk.

In another study, Duan et al. [17] applied AI to assess morphological changes in retinal microvasculature and foveal avascular zone (FAZ) using optical coherence tomography angiography (OCTA), including three vascular parameters and four FAZ-related parameters. This investigation demonstrated that retinal microvascular and macular morphology exhibited damage patterns specific to different subtypes of ischemic stroke.

#### 3.1.2. Prediction of Additional Risk Factors for Stratification

Given the anatomical and physiological similarity between cerebral and coronary circulation [54], non-invasive and accessible retinal abnormalities can directly reflect systemic vascular properties. Thus, in addition to demographic data including age and gender, coronary artery calcium (CAC) and carotid artery atherosclerosis are also frequently used to evaluate CVD risk index.

Poplin et al. [18] proposed a deep-learning model to extract and quantify multiple cardiovascular risk factors from retinal images. The factors used by this model included age, gender, smoking status, HbA1c, and systolic blood pressure, all of which were considered essential for predicting incident CVD.

Retinal age has also been used as a proxy for predisposition to CVD. For example, Nusinovici et al. [19] evaluated the retina-based biological age (RetiAGE) based on a deep learning approach. They reported that RetiAGE, independently of phenotypic biomarkers (e.g., creatinine, glucose, C-reactive protein, etc.) and chronological age, indicated the potential presence of CVD. Moreover, retinal age prediction based on fundus images was also investigated by Zhu et al., who found that the retinal age gap (i.e., predicted retinal age minus chronological age) was significantly associated with arterial stiffness index, incident CVD events [56], and stroke [57].

Many investigations have suggested that retinal image-based deep learning may be a beneficial alternative measure of CAC. For instance, a study conducted by Son et al. [20] demonstrated a high accumulation of CAC. This accumulation was highly correlated with CVD and could be recognized by deep learning algorithms. In another study, Rim et al. [21] developed a three-tier CVD risk stratification system based on Reti-CVD, which showed comparable effectiveness to traditional CT scan measurement in predicting future CVD risk. Furthermore, the authors validated the decent capabilities of Reti-CVD to serve as a risk enhancer tool to predict CVD events using UK biobank data [22].

Using retinal fundus images, Chang et al. [23] predicted atherosclerosis using a similar image analysis strategy. However, the low specificity (i.e., 0.404) suggested that the current algorithm was not suitable for the specific detection of atherosclerosis. Furthermore, they conducted a retrospective cohort analysis to prove that the deep-learning funduscopic atherosclerosis score (DL-FAS), as an independent predictor of CVD mortality, added predictive value over the standard Framingham risk score.

#### 3.1.3. Prediction of Major Cardiocerebral Vascular Events

Previous studies have consistently suggested that major CVD events are predictable using AI algorithms. The aforementioned model by Poplin et al. [18] simultaneously predicted major adverse cardiac events (MACE) based on specific risk factors and obtained an area under the curve (AUC) value of 0.70 (0.65–0.74). Nevertheless, the retinal fundus algorithm did not have an additive effect on the predictive performance of SCORE risk calculation for MACE. In the hybrid system proposed by Diaz-Pinto et al. [24], retinal images and relevant clinical metadata were analyzed together to estimate cardiac indices—including left ventricular mass and left ventricular end-diastolic volume—and to predict incident myocardial infarction; this model showed an AUC of 0.80. In another study, a multimodal deep learning system was proposed by Lee et al. [25] to predict current CVD and assess its performance against a clinical risk factors data-input model simultaneously. The authors found that the integrated network, which combined two networks using two different modalities and made full use of clinical metadata and fundus images, achieved the best performance, with an AUC of 0.872 in the external validation set. Given these results, many patients at high risk of future CVD events may benefit from available retinal image-based AI technology soon.

### 3.2. Detection of Neurodegenerative Diseases and Psychiatric Disorders

As an extension of the cerebrum, the anomalies of vasculature and neuronal structures in the retina are highly associated with cerebral damage. The deposition of Aβ and Tau proteins and the thinning of the nerve fiber layer may be observable in the retina [58]. Recently, several AI algorithms have been generated to forecast neuro-degenerative diseases, including Alzheimer’s disease (AD) [26,27,28,29,30,59,60], Parkinson’s disease (PD) [26,31,32], and Multiple Sclerosis (MS) [33,34]. 

Given that dynamic changes in retinal inner layer thickness can be distinctly revealed by OCT, Nunes et al. [26] built OCT imaging-based machine learning models to detect AD and PD respectively, according to texture analysis. Texture metrics demonstrate the multi-dimensionality of structural arrangements, but complex preprocess can undermine clinical application while improving discrimination. Wang et al. found that macular thickness and volume were in parallel with AD severity [28]. Furthermore, they also successfully demonstrated the feasibility of macular vascular density as a predictive marker for AD [27]. Driven by recent advances in deep learning algorithms, Xie et al. [30] proposed an automatic approach to predicting AD by extracting retinal parameters from OCTA images. However, their results conflicted with previous reports due to the interference of various potential confounds, including vague selection criteria, variation in image quality, differences in equipment, and limited datasets. Despite these confounds, the authors confirmed that microvascular impairment occurred in the retina, which is consistent with the classical AD pathogenesis. In addition, poor equipment reliability [61] seems to be the greatest obstacle to aiding clinical ancillary diagnosis.

Recently, data-rich fundus photography has attracted more attention. For instance, one study by Tian et al. [35] used color fundus photography (CFP) to predict AD, yielding a classification accuracy of 0.82. Moreover, a deep learning model developed by Cheung et al. [29] which inputs retinal images alone was also quite accurate, with a measured accuracy of 83.6%.

With the advances of artificial intelligence and merging eye-tracking technology [62], ophthalmic photography has been increasingly used for the prediction of psychiatric disorders such as schizophrenia (SCZ) [36,63] and autism spectrum disorders [64,65]. For example, a deep learning algorithm developed by Appaji et al. [36] achieved an AUC of 0.98 for classifying SCZ. This result suggests the great potential utility of fundus images in the diagnosis of psychiatric disorders as these protocols improve.

### 3.3. Detection of Chronic Kidney Disease and Renal Function

Homology in microvascular structure between the eye and the kidney suggests that incipient kidney diseases and renal function may be diagnosed using noninvasive ocular images [12].

Sabanayagam et al. [37] developed a deep learning algorithm to detect chronic kidney disease (CKD) from retinal images, and assessment of this model showed that it achieved both good performance and favorable generalization. Although slightly inferior to the combination model that incorporates clinical metadata of risk factors, it implied that accessible retinal photography can be used independently for CKD screening even without detailed medical history acquisition. In one interesting study, Zhang et al. [38] further validated the potential value of an AI-derived image analysis system designed to predict the development of CKD in longitudinal cohorts. Risk stratification performance and the prognostic accuracy of the AI system were both positively assessed, and this model showed an AUC of 0.771 (0.677–0.840) for predicting the onset of CKD in four-year longitudinal data. To explore more predictive biomarkers, Zhang et al. [39] conducted a prospective cohort study to assess the association between the retinal age gap (defined above) and incident kidney failure. Although a significant relationship was revealed, various confounding factors and selection bias were imputed for the limited generalizability of this model.

Additionally, deterioration of renal function reflects a propensity to advance to chronic kidney disease. Rim et al. [40] attempted to use retinal photographs to predict the relative abundances of a series of systemic biomarkers, including creatinine, whose dynamic fluctuations are directly related to renal function. However, this analysis showed relatively poor predictive performance in the European external test. As a result, more comprehensive work is required to explore more precise predictive factors for incident CKD.

### 3.4. Detection of Hematological Diseases

Anemia, as the most common hematological disease, has been generally predicted with ophthalmic imaging-based deep learning algorithms based on conjunctival signs [41,42,66], and subtle alterations in the retina [43,44,45,46]. Chen et al. [42] proposed an automatic framework that integrated semantic segmentation and a convolutional neural network based on prior causality to predict eyelid Hb, and this model showed an R^2^ value of as high as 0.512. In retrospect, the predictive precision of conjunctival imaging-based models encounters several key restrictions, including acquisition criteria, image extraction, and algorithm selection. In contrast, Mitani et al. implemented a promising screen for anemia with a CFP-based deep learning algorithm [44]. Afterward, Zhao et al. [45] used UWF images as input for a deep learning model, and this realized an excellent AUC of 0.93 (0.92–0.95), implying that the retinal surrounding area can also contribute to the precise prediction of anemia. Recent literature has found that capillary plexus density and retinal microvascular perfusion decrease in OCTA in patients with anemia [67]. Accordingly, a lightweight network [46] proposed based on retinal vessel OCT images achieved decent performance (i.e., 0.99 AUC); however, this model was confined to the diminutive dataset and was not validated externally. Moreover, Wu et al. [43] constructed a model to identify pregnant patients with anemia by combining metadata with quantitative OCTA metrics. They found their combined model reached the best performance, realizing an AUC of 0.874 (0.835 to 0.914).

### 3.5. Detection of Other Systemic Diseases

#### 3.5.1. Detection of Hepatobiliary Diseases

Disturbances in glucolipid metabolism and the accumulation of abnormal metabolic toxicants can disrupt a variety of physiological processes. This can lead to many corresponding ocular abnormalities, [68] which frequently exhibit as non-specific and indecipherable. Xiao et al. [47]. developed seven deep-learning models to detect six major hepatobiliary diseases using slit-lamp and fundus images, respectively. By comparison, the identification of liver cancer and liver cirrhosis achieved better performance using slit-lamp images than using fundus images. This unanticipated finding implies that imperceptible alterations exist in the retina of patients with hepatobiliary diseases.

#### 3.5.2. Detection of Lung Neoplasms

Lung cancer, the leading cause of cancer deaths, accounts for 18.0% of all deaths from cancer worldwide [69]. Recently, Huang et al. [48] put forward a convenient and non-invasive approach to detecting lung neoplasms. They trained a multi-instance learning model to distinguish benign from malignant pulmonary nodules using scleral images. Despite data imbalances caused by subject selection bias, this binary classification task was completed well, eventually with an average AUC of 0.897. Prior to this, another study used scleral images to detect polycystic ovary syndrome (PCOS) with a promising result [49]. Taken together, these studies show that the scleral image-based deep learning model possesses tremendous application prospects for clinical screening.

#### 3.5.3. Detection of Sarcopenia

Sarcopenia is an age-related skeletal muscle disorder where the accelerated loss of muscle mass and functional decline are the predominant symptoms [70]. The deep learning algorithm proposed by Rim et al. [40] was developed to quantify body muscle mass from retinal photographs to evaluate the risk of sarcopenia onset. This model performed satisfactorily, achieving an average absolute error of 6.09 kg (R^2^ = 0.33), which is comparable to bioelectrical impedance analysis. Kim et al. [50] proposed a concept of predicting sarcopenia risk with machine learning-based oculomics incorporating ocular measurements and demographic factors, which achieved impressive performance. However, due to restricted access to the raw photographs in the database, the potential information contained in fundus photographs was not sufficiently exploited. 

## 4. Discussion

Increasing healthcare expenditures and deteriorating health conditions prompt the consideration of trade-off maximization. That is, the financial burden imposed on society due to rapid increases in medical expenses is causing healthcare providers to identify newer, cheaper diagnostic methods, especially for areas in which medical resources are scarce [71]. AI-based ocular image analysis, which is non-invasive, cost-effective, and repeatable, is expected to serve as an alternative medical screening strategy.

### 4.1. Dilemmas and Trends

Novel AI-based approaches to disease prediction can tie seemingly distant disciplines of medicine together. However, despite the current promising achievements presented above, the clinical translation of ocular image-based AI for the prediction of systemic diseases still faces formidable obstacles, including bias, data accessibility, privacy and security [72], and lack of transparency and generalizability [2].

First, interdisciplinary collaboration is necessary to coordinate and thereby improve the prediction of systemic diseases via ocular imaging-based AI. This is critical, since input images and target diseases may need to be collected separately and consciously. In addition, the absence of uniform standards for data compilation and storage restricts the circulation and sharing of medical data. This “information islands” dilemma urgently requires the establishment of a standardized data acquisition protocol across academic disciplines and nations to consolidate the foundational datasets used to develop and test AI applications [73].

Second, inadequate interpretability ascribed to the “black box” nature of AI methods is the most fundamental constraint for the clinical implementation of ocular im-age-based AI models [74,75]. Partial systemic diseases with a well-established pathophysiologic link with the retina have been confirmed by several studies, such as CVD and dementia. In addition to these two cases, novel applications for the detection of hepatobiliary diseases [47], lung neoplasms [48], and PCOS [49] by analyzing subtle variations in external eye images remain incomprehensible, even with the assistance of interpretability analysis. In high-stakes scenarios such as medical care, any suboptimal treatment or misdiagnosis attributable to the unexplainability may generate disastrous clinical, ethical, and legal consequences [76]. Accordingly, it is pivotal for the enhancement of algorithm theoretical support to figure out the underlying pathophysiological mechanisms of the alternations. Furthermore, further prospective studies based on larger sample sizes are required and must have suitable control of bias and confounds. Moreover, for the full use of information, the integration of clinical data and ocular images is encouraged for the prediction of systemic diseases. Finally, to increase public credibility, rigorous and thorough validation should be prioritized before clinical deployment [77,78].

Third, comorbidity introduces significant clinical complexity and is another inevitable challenge to overcome. The prediction of systemic disease from ocular images will face interference from other pathological factors which can cause specific ocular abnormalities simultaneously, including diabetes [79], hypertension [80], and hyperthyroidism [81]. The concurrent existence of multiple retinal pathologies in a fundus image may also undermine the predictive performance of AI models. Therefore, multimodal and multitask learning architecture should be encouraged to isolate the alternations caused by confounding factors. Furthermore, the integrated input of clinical metadata and ocular images, in most cases, may improve predictive accuracy [25].

Fourth, the current applications of ocular image-based AI in detecting systemic diseases are dominated by retrospective studies. However, prospective research involving preclinical and clinical studies with longitudinal images and data chains is required to mitigate various kinds of bias. For systemic disease, non-ocular systemic diseases in particular, the ophthalmologic examination is not a standard required procedure. Hence, it is critical to discreetly weigh it against routine standard tests and determine whether there may be incremental value for target disease screening.

Finally, limited data accessibility is the most common problem encountered during the development of related AI models. Insufficient matching information of systemic disease-related data and objective ophthalmic examination data restricts the effective individual development of models. While in publicly accessible datasets, the heterogeneity of composition always leads to biased predictions and poor generalizability. This manifests as poor performance in complex real-world scenarios. For example, when the algorithms of Rim et al. [40], which were trained on data from Asian participants were validated in the UK Biobank dataset, poorer performances were observed; this may be attributed to the discrepancies caused by ethnicity [82]. We have to admit that the composition of datasets involving age, gender, health condition [18], and ethnicity must be considered during algorithms development and performance evaluation.

### 4.2. Strength and Limitations

This review focuses on current applications of AI related to the prediction of systemic diseases from multimodal ocular images. This novel diagnostic solution is a concrete demonstration of a global and holistic view of AI applications. We believe that it will provide a new perspective for resolving complex medical issues. However, we also admit that many limitations still exist. Given the constraints of the search strategy and search date, some cutting-edge research was not included because they were not peer-reviewed or because no full text was available. In addition, the exclusion of articles published in languages other than English may also result in the omission of useful results. In addition, the primary study searching and screening phase conducted by a single assessor may lead to subjective bias and/or unknown errors. Moreover, the field of AI is evolving rapidly, and future reviews are preferred to provide a more comprehensive and objective presentation of published clinical AI applications. Furthermore, we will propose more feasible and higher-quality recommendations to guide future work by tracking rapidly changing trends.

## 5. Conclusions

Ophthalmic images contain large amounts of imperceptible information which can be easily decoded by AI. For this reason, ocular image-based AI models for predicting systemic disease have great potential for clinical screening. Nevertheless, the “black box” nature of AI makes it crucial to conduct further validation research to improve the reliability and efficacy of the clinical application. Given the interdisciplinary nature of this technology, collaborative efforts should be stressed to incorporate advanced technology into practice and to improve medical care comprehensively. Despite limitations and challenges, AI is likely to cast an indispensable role in the medical field in the future.

## Figures and Tables

**Figure 1 healthcare-11-01739-f001:**
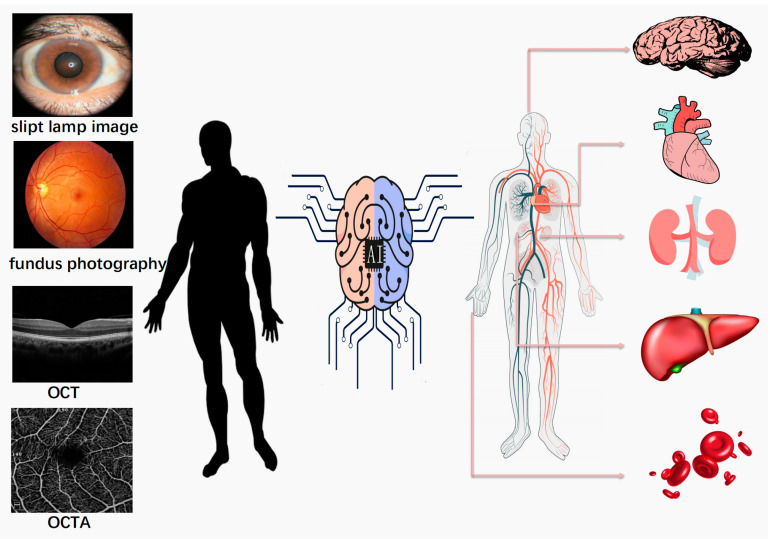
Overview of predictable systemic diseases capable of being assessed using primary ophthalmic imaging modalities and AI technology. These ocular images were derived from the authors’ research database.

**Figure 2 healthcare-11-01739-f002:**
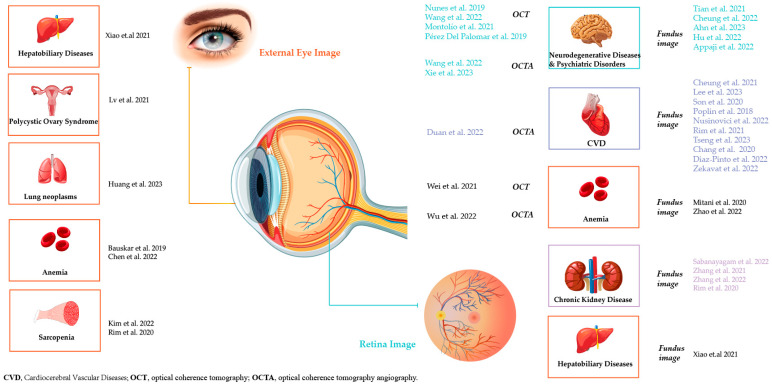
Classification of the included articles according to the image modalities used and the target systemic diseases [15,16,17,18,19,20,21,22,23,24,25,26,27,28,29,30,31,32,33,34,35,36,37,38,39,40,41,42,43,44,45,46,47,48,49,50].

**Table 1 healthcare-11-01739-t001:** AI-related Studies for Cardiocerebral Vascular Diseases Prediction.

Author, Year	Target Disease	Input	Architectures	Training/Testing Dataset	External Validation	Recruitment	Task ^3^	Outcome	Output	Results ^1^
Cheung et al., 2021 [15]	CVD	Fundus images	CNN	SEED: 5309 images	10 external datasets: 5636 images	Retrospective	Prediction	Retinal-vessel morphological parameters	Regression	Related ^2^
Lee et al., 2023 [25]	CVD	Fundus images	CNN, DNN	Samsung Medical Center: 3518 images for training, 2954 images for internal validation	UK Biobank: 11,298 images	Retrospective	Prediction	Clinical diagnosis	Binary	AUC = 0.872
Son et al., 2020 [20]	CVD	Fundus images	Inception-v3	Seoul National University Bundang Hospital: 44,184 images for training, and 5-fold validation	N/A	Retrospective	Prediction	Coronary artery calcium score	Binary	AUC = 0.832
Poplin et al., 2018 [18]	MACE	Fundus images	Inception-v3	UK Biobank: 96,082 images for training, 24,008 images for validation; EyePACS: 1,682,938 images for training, 1958 images for validation	N/A	Retrospective	Prediction	Clinical diagnosis	Binary	AUC = 0.70
Nusinovici et al., 2022 [19]	CVD	Fundus images	RetiAGE	Korean Health Screening study: 129,236 images for training	UK Biobank: 56,301 patients.	Retrospective	Prediction	Biological age	Regression	Related
Rim et al., 2021 [21]	CACS	Fundus images	RetiCAC	South Korean health screening centre1: 5590 images for internal testing	South Korean health screening centre2: 18,920 images; CMERC-HI: 1054 images; SEED: 17,102 images; UK Biobank: 93,358 images	Retrospective	Prediction	Coronary artery calcium score	Binary	AUC = 0.742
Tseng et al., 2023 [22]	CACS	Fundus images	Deep learning	UK Biobank: 48,260 patients	N/A	Retrospective	Prediction	Reti-CVD score	Regression	Related
Chang et al., 2020 [23]	CVD	Fundus images	DL-FAS	Seoul National University Hospital: 15,408 images for training	N/A	Retrospective	Prediction	Atherosclerosis Score	Binary	AUC = 0.713
Duan et al., 2022 [17]	Ischemic Stroke	OCTA	OCTA-Net	Peking University Third Hospital: 60 patients	N/A	Retrospective	Identification	Retinal-vessel morphological parameters	Regression	Related
Diaz-Pinto et al., 2022 [24]	Myocardial Infarction	Fundus images	ResNet50	UK Biobank: 87,476 participants	N/A	Retrospective	Prediction	Cardiac indices	Binary	AUC = 0.80
Zekavat et al., 2022 [16]	Congestive heart failure	Fundus images	CNN	UK Biobank: 97,895 images	N/A	Retrospective	Prediction	Retinal-vessel morphological parameters	Regression	Related

^1^ Only the best performance is presented when there was more than one model. ^2^ Related means there is a clear association between variables obtained from ocular images and target disease without accessible R^2^. ^3^ Task includes identification and prediction. The identification task means identifying the current state of the target systemic disease from ocular images in a cross-sectional study, while the prediction task refers to predicting future events or incident risk from ocular images in a longitudinal cohort. SIVA-DLS, Singapore I Vessel Assessment-Deep Learning System; SEED, Singapore Epidemiology of Eye Diseases; N/A, not available; CMERC-HI, Cardiovascular and Metabolic Disease Etiology Research Center-High Risk Cohort; CVD, cardiovascular diseases; CNN, convolutional neural network; DNN, deep neural network; MACE, major adverse cardiovascular events; EyePACS, Eye Picture Archive Communication System; RetiAGE, retina-based biological age; RetiCAC, retina-based coronary artery calcium; OCTA, optical coherence tomography angiography; AUC, the area under the curve.

**Table 2 healthcare-11-01739-t002:** AI-related Studies for Neurodegenerative Diseases and Psychiatric Disorders.

Author, Year	Target Disease	Input	Architectures	Training/Testing Dataset	External Validation	Recruitment	Task ^3^	Outcome	Output	Results ^1^
Nunes et al., 2019 [26]	AD PD	OCT	SVM	University of Coimbra, Portugal: 20 AD patients, 28 PD patients, 27 HC	N/A	Retrospective	Identification	Clinical diagnosis	Ternary	ACC = 0.822
Wang et al., 2022 [28]	AD	OCT	XGBoost	Xiangya Hospital:159 AD patients, 299 HC	N/A	Prospective	Identification	Clinical diagnosis	Binary	ACC = 0.74
Wang et al., 2022 [27]	AD	OCTA	Adaptive boosting	Xiangya Hospital: 77 AD patients, 145 HC	N/A	Prospective	Identification	Clinical diagnosis	Binary	AUC = 0.73
Xie et al., 2023 [30]	AD	OCTA	OCTA-Net, FAZ-Net	Affiliated People’s Hospital of Ningbo University, Peking University Hospital: 55 AD patients, 41 MCI patients, 62 HC	N/A	Prospective	Identification	Retinal-vessel morphological parameters	Regression	Related ^2^
Tian et al., 2021 [35]	AD	Fundus images	Multi-stage machine learning	UK Biobank: 122 images AD patients and 52,492 images HC, for training and 5-fold cross-validation	N/A	Retrospective	Identification	Clinical diagnosis	Binary	ACC = 0.8244
Cheung et al., 2022 [29]	AD	Fundus images	EfficientNet-b2	Eight centers in four countries: 12 949 images for training and testing, 648 AD patients, 3240 HC	N/A	Retrospective	Identification	Clinical diagnosis	Binary	AUC = 0.73
Ahn et al., 2023 [32]	PD	Fundus images	ResNet-18, Fully connected neural network	Kangbuk Samsung Hospital, Seoul National University Hospital and Yeungnam University Hospital: 266 PD patients, 349 HC	N/A	Prospective	Identification	Clinical diagnosis	Binary	AUC = 0.67
Hu et al., 2022 [31]	PD	Fundus images	N/A	UK Biobank: 19,200 images	N/A	Retrospective	Prediction	Retinal age gap	Binary	AUC = 0.708
Montolío et al., 2021 [34]	MS	OCT	LSTM recurrent neural network	Miguel Servet University Hospital: 108 MS patients, 104 HC	N/A	Prospective	Prediction and Identification	Clinical diagnosis	Binary	AUC = 0.8165
Pérez Del Palomar et al., 2019 [33]	MS	OCT	Random Forest with Adaboost	Miguel Servet University Hospital: 80 MS patients, 180 HC	N/A	Prospective	Identification	Clinical diagnosis	Binary	AUC = 0.998
Appaji et al., 2022 [36]	SCZ	Fundus images	CNN	NIMHANS: 139 SCZ patients, 188 HC for training; 33 patients, 23 HC for validation; 17 patients, 13 HC for Testing	N/A	Prospective	Identification	Clinical diagnosis	Binary	AUC = 0.98

^1^ Only the best performance is presented when there was more than one model. ^2^ Related means there is a clear association between variables obtained from ocular images and target disease without accessible R^2^. ^3^ Task includes identification and prediction. The identification task means identifying the current state of the target systemic disease from ocular images in a cross-sectional study, while the prediction task refers to predicting future events or incident risk from ocular images in a longitudinal cohort. SVM, support vector machine; OCT, optical coherence tomography; OCTA, optical coherence tomography angiography; N/A, not available; ACC, accuracy; AUC, the area under the curve; MCI, mild cognitive impairment; HC: healthy control; SCZ, schizophrenia; AD, Alzheimer disease; PD, Parkinson Disease; MS, multiple sclerosis; CNN, convolutional neural network; NIMHANS, National Institute of Mental Health and Neurosciences.

**Table 3 healthcare-11-01739-t003:** AI-related Studies for Anemia and Chronic Kidney Disease.

Author, Year	Target Disease	Input	Architectures	Training/Testing Dataset	External Validation	Recruitment	Task ^3^	Outcome	Output	Results ^1^
Sabanayagam et al., 2022 [37]	CKD	Fundus images	CondenseNet	SEED: 5188 patients for training, 1297 patients for validation	SP2:3735 patients BES: 1538 patients	Retrospective	Identification	Laboratory index(eGFR) and Clinical diagnosis	Binary	AUC = 0.835
Zhang et al., 2021 [38]	CKD, Early CKD	Fundus images	ResNet-50	CC-FII data: 60,244 images for training, 8614 images for tuning, and 17,454 images for internal testing	CC-Fll data: 16,118 images; COACS: 6162 images	Prospective	Prediction and Identification	Laboratory index(eGFR) and Clinical diagnosis	Binary Binary	AUC = 0.885 AUC = 0.834
Zhang et al., 2022 [39]	Kidney Failure	Fundus images	N/A	UK Biobank: 35,864 participants	N/A	Retrospective	Prediction	Retinal age gap	Regression	Related ^2^
Bauskar et al., 2019 [41]	Anemia	Conjunctiva images	Modified SVM	48 anemia patients, 51 HC for training, and k-Fold cross-validation	N/A	Prospective	Identification	Clinical diagnosis	Binary	AUC = 0.93
Chen et al., 2022 [42]	Anemia	Conjunctiva images	Mask RCNN, MobileNetv3	Southwest Hospital: 1065 patients	N/A	Retrospective	Identification	Laboratory index (Hb) and Clinical diagnosis	Regression	R^2^ =0.512
Mitani et al., 2020 [44]	Anemia	Fundus images	Inception-v4	UK Biobank: 7163 participants, 70% for training, 10% for tuning, and 20% for validation	N/A	Retrospective	Identification	Laboratory index (Hb) and Clinical diagnosis	Binary	AUC = 0.87
Zhao et al., 2022 [45]	Anemia	Fundus images	ASModel_UWF, ASModel_Cropped UWF	Peking Union Medical College Hospital: 9221 images for training, 577 images for validation, 1730 images for testing	N/A	Prospective	Identification	Laboratory index (Hb) and Clinical diagnosis	Binary	AUC = 0.93
Wei et al., 2021 [46]	Anemia	OCT	AneNet	the Second Xiangya Hospital: 17 anemia patients, 221 images; 13 HC, 207 images for training, 5-fold cross-validation	N/A	Retrospective	Identification	Clinical diagnosis	Binary	AUC = 0.9983
Wu et al., 2022 [43]	Anemia	OCTA	N/A	Zhongshan Ophthalmic Centre, First Affiliated Hospital of Sun Yat-sen University: 99 patients, 184 HC	N/A	Prospective	Identification	Clinical diagnosis	Binary	AUC = 0.874

^1^ Only the best performance is presented when there was more than one model. ^2^ Related means there is a clear association between variables obtained from ocular images and target disease without accessible R^2^. ^3^ Task includes identification and prediction. The identification task means identifying the current state of the target systemic disease from ocular images in a cross-sectional study, while the prediction task refers to predicting future events or incident risk from ocular images in a longitudinal cohort. CKD, chronic kidney disease; SEED, Singapore Epidemiology of Eye Diseases; eGFR, estimated glomerular filtration rate; Hb, Hemoglobin; CC-FII, China Consortium of Fundus Image Investigation; SP2, Singapore Prospective Study Program; BES, Beijing Eye Study; COACS, China suboptimal health cohort study; N/A, not available; AUC, the area under the curve; UWF, ultra-wide-field; OCT, optical coherence tomography; OCTA, optical coherence tomography angiography; SVM, support vector machine.

**Table 4 healthcare-11-01739-t004:** AI-related Studies for Other Systemic Diseases.

Author, Year	Target Disease	Input	Architectures	Training/Testing Dataset	External Validation	Recruitment	Task ^2^	Outcome	Output	Results ^1^
Xiao et al., 2021 [47]	Liver cancer, Liver cirrhosis, Chronic viral hepatitis, NAFLD Cholelithiasis, Hepatic cyst	Fundus images, Slit-lamp images	ResNet-101	1252 participants, 2481 slit-lamp images, 1989 retinal images; 75% for training,20% for tuning	1069 slit-lamp images, 800 retinal images	Prospective	Identification	Clinical diagnosis		Slit-lamp;
										Fundus:
									Binary	AUC = 0.93; 0.84
									Binary	AUC = 0.90; 0.83
									Binary	AUC = 0.69; 0.62
									Binary	AUC = 0.63; 0.70
									Binary	AUC = 0.58; 0.68
									Binary	AUC = 0.66; 0.69
Huang et al., 2023 [48]	Lung neoplasms	Scleral images	U-Net, Resnet-18, MIL model	Emergency General Hospital: 950 scleral images	N/A	Prospective	Identification	Clinical diagnosis	Binary	AUC = 0.897
Lv et al., 2021 [49]	PCOS	Scleral images	U-Net, Resnet-18, MIL model	Peking University Third Hospital: 4608 images for training, 1160 images for testing, 5-fold cross-validation	N/A	Retrospective	Identification	Clinical diagnosis	Binary	AUC = 0.979
Rim et al., 2020 [40]	CKD Sarcopenia	Fundus images	VGG-16	Seven Asian and European cohorts: 86,994 images for training, 21,698 for testing	Health screening center affiliated with the Severance Gangnam Hospital: 9324 images; BES: 4234 images; SEED: 63,275 images; UK Biobank: 50,732 images	Retrospective	Identification	Laboratory index (Creatinine) Body muscle mass	Regression Regression	R^2^ = 0.12 R^2^ = 0.36
Kim et al., 2022 [50]	Sarcopenia	Fundus images, Slit-lamp images	XGBoost	KNHANES: 8092 participants	N/A	Retrospective	Identification	Clinical diagnosis	Binary	Male: AUC = 0.75 Female: AUC = 0.785

^1^ Only the best performance is presented when there was more than one model. ^2^ Task includes identification and prediction. The identification task means identifying the current state of the target systemic disease from ocular images in a cross-sectional study, while the prediction task refers to predicting future events or incident risk from ocular images in a longitudinal cohort. CVH, Chronic viral hepatitis; NAFLD, non-alcoholic fatty liver disease; AUC, the area under the curve; N/A, not available; CKD, chronic kidney disease; KNHANES, Korean National Health and Nutrition Examination Survey; PCOS, Polycystic Ovary Syndrome.

## Data Availability

Not applicable.

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
