# Peer review of "Diagnosing Systemic Disorders with AI Algorithms Based on Ocular Images"

_healthcare, 2023, doi:10.3390/healthcare11121739_

Round 1

Reviewer 1 Report

This narrative review overviews the potential of ophthalmic imaging technologies with regard to screening and detection of systemic diseases with the help of artificial intelligence. While such an overview is interesting in this rapidly evolving field, some clarifications need to be made in order to warrant for peer-review publication

The material and methods section defines search terms, but not inclusion and exclusion criteria. This should be refined, preferably according to the PRISMA guidelines. 

More detail in the predicted outcome and the dataset is needed in Tables 1-4, especially dataset size, bias, and if the outcome was a clinical or a laboratory outcome, and if the study is predicting future events or detecting current state of disease.

Generally, this narrative review is easily read and giving a wide overview. However, the authors miss the chance of providing critical discussion or innovative inputs with regard to AI specific challenges and future exams. 

Reviewer 2 Report

1. The main question is the low significance. Many reviews about AI, systemic disorders, and ocular images were published, like "Detection of Systemic Diseases From Ocular Images Using Artificial Intelligence: A Systematic Review" and others. What is your difference?

2. Many language and grammar errors in the text.

Many language and grammar errors in the text.

Reviewer 3 Report

The manuscript is with merit, but the authors should address the following comments before publication can be considered.

The English grammar and style should be revised and improved in all the manuscript.

In the introductions the authors should provide a definition of AI

Methods: the authors should add the dates in which the literature search was performed

The authors should make sure that all the abbreviations used in the manuscript have a corresponding explanation and that after one abbreviation is introduced and explained, only the abbreviation is used in the rest of the manuscript.

The authors should add a "limitations" section of their review before the conclusions

The authors should indicated the # of the figure 1 they are referring to in the body of the manuscript (they just mentioned "FIGURE")

Moderate editing of English language

Round 2

Reviewer 2 Report

We thank the authors for clarifying and improving. But the imitation trace is strong, and the authors cannot address my concerns sufficiently to make this manuscript suitable for publication.

Improve a lot, requiring minor editing.
